# COVID-19 and Female Genital Mutilation/ Cutting and child marriage: An online multi-country cross sectional survey

Shania Pande[1], Simukai Shamu[2,3], Amr Abdelhamed[4], James Munyao Kingoo[5], Sarah Van de Velde[6], Marleen Temmerman[7], Tammary Esho[8], Samuel Kimani[9], Joyce Omwoha[10], Eneyi E. Kpokiri[11]*, Joseph D. Tucker[11,12]*

1 Faculty of Public Health and Policy, London School of Hygiene and Tropical Medicine, London, United Kingdom, 2 Health Systems Strengthening, Foundation for Professional Development, Pretoria, South Africa, 3 School of Public Health, University of Witwatersrand, Johannesburg, South Africa, 4 Department of Dermatology, Venereology & Andrology, Sohag University, Sohag, Egypt, 5 Department of Biochemistry and Biotechnology, School of Life and Biological Sciences, Technical University of Kenya, Nairobi, Kenya, 6 Centre for Population, Family, and Health, Department of Sociology, University of Antwerp, Antwerp, Belgium, 7 Aga Khan University Hospital, Nairobi, Kenya, 8 Amref International University, Nairobi, Kenya, 9 University of Nairobi, Nairobi, Kenya, 10 Technical University of Kenya, Nairobi, Kenya, 11 Faculty of Infectious and Tropical Diseases, London School of Hygiene and Tropical Medicine, London, United Kingdom, 12 School of Medicine, University of North Carolina at Chapel Hill School of Medicine, Chapel Hill, North Carolina, United States of America

* Eneyi.Kpokiri@lshtm.ac.uk (EEK); jdtucker@med.unc.edu (JDT)

**Data Availability Statement:** All relevant data are within the paper and its Supporting Information files.

## Abstract

Female Genital Mutilation/Cutting (FGM/C) and child marriage are prevalent in many countries in Asia and Africa. These practices are a violation of human rights and have significant impacts on the physical and mental well-being of those affected. COVID-19 restrictions such as lockdowns and closure of schools may have influenced the occurrence of FGM/C and child marriage. This analysis reported on the impact of these restrictions on FGM/C and child marriage. The International Sexual Health And REproductive Health (I-SHARE) research team organised a multi-country online survey. Sampling methods included convenience samples, online panels, and population-representative samples. Data collected included the impact of COVID-19 restrictions on the occurrence, intention to practice and change in plans to organise FGM/C and child marriage. Data were analysed from 14 countries that reported on FGM/C and child marriage using basic descriptive statistics. Given it was an online survey, we had more responses from urban areas. Among the 22,724 overall participants, 8,829 participants (38.9%) responded to the survey items on FGM/C and child marriage and were included in this analysis. 249 (3.4%) participants stated that FGM/C occurred in their community during COVID-19. Out of this, COVID-19 affected the plans of 26 (20%) participants intending to organise circumcision and 15% of participants planned to organise FGM/C earlier. People with a worry about finances during COVID-19 were more likely to have an earlier plan to organise FGM/C during COVID-19. In total, 1,429 (13%) participants reported that child marriage occurred in their community. The pandemic affected plans of 52 (13%) participants intending to arrange child marriage and 7.7% (29/384) participants expressed intent to arrange the marriage sooner than planned. People with financial

**Funding:** The author(s) received no specific funding for this work.

**Competing interests:** The authors have declared that no competing interests exist.

insecurities during COVID-19 were more likely to arrange a child marriage earlier. Thus, our study found that the pandemic impacted plans related to FGM/C and child marriage practices, resulting in many carrying out the practices sooner or later than initially planned.

## Introduction

Female genital mutilation/cutting (FGM/C) is any procedure that involves the partial or complete removal of the female genitalia or injury to the female reproductive organs for non-medical reasons [1]. According to a UNICEF report, more than 200 million girls have undergone FGM/C globally [2]. FGM/C has been reported in 30 countries, mainly in Africa, the Middle East and Asia [3]. The practice is deeply rooted in local patriarchal cultures and many people feel that circumcision is a traditional part of a girl's entry into womanhood [4]. Not only is FGM/C a violation of a girl's rights, but it has also been linked to numerous health risks, such as increased risk of reproductive anomalies, urinary tract infections, HIV, sexual dysfunction, and menstrual problems [5, 6]. It is also associated with mental health consequences such as depression, anxiety, and stress [7]. Furthermore, medical procedures associated with treating the heath impacts of FGM/C carry substantial economic costs to the world, estimated at 1.4 billion USD per year as of 2020 [1].

FGM/C may be performed in preparation for child marriage; however, this has reduced in past few years [8]. In some settings, it is performed to prepare young girls for sexual activity and pregnancy. Child marriage refers to formal or informal unions wherein one or more of the people to be married are under the age of 18 years [9]. Globally, the prevalence of child marriage in girls is six times as compared to boys [9]. According to UNICEF, over 12 million child marriages take place across the world every year [10, 11]. Child marriage is more common among poor families [12]. In cultures where a girl's education is limited, many parents view marriage as an opportunity to secure a more stable economic future for their daughter and her family. Child marriage can have a significant effect on a girl's physical, mental and emotional health. For example, child marriage is associated with early pregnancy, malnourishment, and stunting [13]. Furthermore, early pregnancy also has other negative socioeconomic impacts, such as the early termination of academic pursuits to support their families. Limiting a child's access to education also limits their economic opportunities and may further exacerbate poverty [14].

Restrictions imposed due to the COVID-19 pandemic such as lockdowns and closures of schools may have exacerbated harmful cultural practices like FGM/C and child marriage. Several reports suggest that COVID-19 may have exacerbated FGM/C [15–17]. The financial implications of the pandemic may have also increased poverty in specific areas and moved ahead plans for FGM/C and child marriage [18, 19]. For example, Bangladesh reported an increased number of child marriages prior to and during COVID-19 [20, 21]. Children from poorer households were found to be the most vulnerable to early marriage [22]. There is a precedent that periods of insecurity and economic instability drive child marriage. For example, during the Ebola outbreak, there was a steep increase in child marriage in Western Africa [23, 24]. Similarly, the conflicts in Syria, Kenya, and Somalia prompted many parents to arrange a marriage for their daughters to older men or men in the military to protect them from being kidnapped or sexually assaulted [25, 26].

This paper reports on the impact of the COVID-19 restrictions on FGM/C and child marriage in 14 countries using online cross-sectional survey data.

## Methods

This study was a multi-country analysis conducted as part of the International Sexual Health And REproductive Health (I-SHARE) Survey. I-SHARE is a consortium of researchers at the Academic Network for Sexual and Reproductive Health and Rights Policy (ANSER) and the London School of Hygiene and Tropical Medicine (LSHTM), along with several partner organisations in all countries where the survey was implemented [27]. Through a series of cross-sectional online surveys, we aimed to investigate the impact of COVID-19 restrictions on different aspects of sexual and reproductive health. The multi-country analysis included data from 30 different high- (HICs), middle- (MICs) and low-income (LICs) countries.

### Data collection

A cross-sectional survey design was used in each country to collect data [27]. Sampling strategies included convenience (twenty-three countries), online panels (six countries), and population-representative (two countries). The data was collected during the initial COVID-19 wave from (July 2020–February 2021) and was accessed for use in this research in June 2021.The lead organisation in each country selected networks to distribute the link to the online survey. The link was disseminated through networks, email listservs, local partner organisations and social media [27]. The survey was self-administered by participants using a tablet, computer or smartphone. Further, the survey was translated into the national language of participating countries. The inclusion criteria were as follows: must be 18 years or older; residing in one of the participating countries; and able to provide online informed consent.

### Ethical approval

Online written and informed consent was obtained from participants. The study received ethical approval from Ghent University and a de-identified version was approved by the University of North Carolina at Chapel Hill. Further, local ethical approvals were also obtained in the thirty countries where the survey was implemented.

### Survey items

The survey instrument was developed based on existing and newly developed items [28]. Paper-based and digital field testing of the survey instrument was performed in local contexts, and the survey instrument was then revised accordingly. The survey items for FGM/C and child marriage were adapted from already validated surveys [29–31]. Participants were asked whether they intended to carry out both practices and if they were aware of the occurrence within their family or communities. An example of a question used for data collection "Is female circumcision practised in your community?", "Before the COVID-19 social distancing measures, did you intend to circumcise your daughter?" and "Did the COVID-19 situation change your plans to circumcise your daughter?" The complete list of survey questions and response options for FGM/C and child marriage can be found in the supporting information (S1 Table). Data on socio-demographics such as age of participants, sex at birth, gender, number of children, area of residence, educational level, economic condition, religion and ethnicity were also collected. Note that the survey items for FGM/C and child marriage were included as optional components of the main survey and 14 countries included these two components in the survey. The survey did not include questions about the type of COVID restrictions.

## Statistical analysis

In this particular study, we analysed data from 12 countries that reported on survey items relating to FGM/C and 14 countries which reported on survey items relating to child marriage. Statistical analysis was performed using Stata (version 16.1). Socio-demographic characteristics of the study population were tabulated and presented using basic descriptive frequencies. Descriptive statistics such as sample sizes and frequencies were used to cross-tabulate the survey items for FGM/C and child marriage by country. Correlation analysis of the socio-demographics such as age of the participant, religion, income, number of children, educational status, and area of residence with the decision to carry out FGM/C or child marriage was performed, and chi-squared values were obtained. The level of statistical significance was p≤0.05.

## Results

### Socio-demographics of the study population

Our analysis used data from 8,829 participants in 14 countries that reported on any of the survey items relating to FGM/C and child marriage. The mean age of participants was 32 years (± 11 years) old; 68.7% (6,062/8,829) were female at birth and 31.2% (2,754/8,829) were male. About 63% (5,642/8,829) lived in HICs, 25.2% (2,226/8829) in upper-middle income countries (UMICs) and 10.8% (961/8,829) in lower-middle income countries (LMICs). Further, 70.2% (6,185/8,828) lived in urban areas while 20% (1,762/8,829) of people lived in rural areas (Table 1).

### FGM/C

Twelve countries reported on the occurrence of FGM/C in their community. In total, 249 (3.4%) out of the total 7,234 participants stated that FGM/C occurred in their community (S1 Table). Countries with a higher occurrence of FGM/C included Kenya (86/249, 34.5%) Singapore (59/249, 23.7%), Nigeria (28/249, 11.2%), Egypt (20/249, 8%) and Portugal (18/249, 6.1%).

About 13.6% (18 out of 132) of participants intended to organise circumcision for their daughter. This included people in Singapore, Egypt, Italy, Kenya, Luxembourg, Nigeria, Australia, Portugal, and Spain. Among this group, 11 out of the 18 participants were from Singapore. When looking at the impact of COVID-19 on plans to organise circumcision, our results found that the pandemic affected the decision of 20% of participants to organise circumcision. Out of these, 19 (14.6%) participants intended to circumcise their daughters sooner than initially planned, while 6 (4.6%) stated later than planned (Fig 1); only one (0.8%) participant had cancelled plans for circumcision.

Univariate associations of the correlates of changing plans to circumcise due to COVID-19 (Table 2) showed people with higher levels of financial worries were more likely to change plans ($X^2$ = 14.03; p = 0.02). About 11.8% of those more worried about their financial situation planning to organise circumcision sooner than planned, while 84.3% did not change plans for circumcision. However, change in economic situation was found to be not significant in changing plans ($X^2$ = 11.58; p = 0.07), as was education level ($X^2$ = 4.77; p = 0.97). No association was found between change in plans and other variables such as area of residence and number of children. Further, about 58% (161/278) of participants felt that COVID-19 put girls at an increased risk of FGM/C (S1 Fig). Note that participant's perception of risk was not found to be associated with the decision to change plans to organise FGM/C.

**Table 1. Socio-demographics of I-SHARE study participants in 14 countries that responded to FGM/C and child marriage survey items, 2020–2021 (N = 8,829).**

| Socio-Demographics | | Frequency (N = 8,829) | Percentage |
|---|---|---|---|
| Age* | 18–25 | 2,315 | 27.0% |
| | 26–40 | 4,226 | 49.2% |
| | 41–50 | 1,200 | 14.0% |
| | 51–60 | 585 | 6.8% |
| | 61–70 | 224 | 2.6% |
| | 71–80 | 27 | 0.3% |
| | 81+ | 7 | 0.1% |
| Sex at Birth | Male | 2,754 | 31.2% |
| | Female | 6,062 | 68.7% |
| | Other | 13 | 0.6% |
| Gender** | Cisgender | 7,338 | 83.4% |
| | Non-cisgender | 1,216 | 13.8% |
| | Other | 28 | 0.3% |
| | NA | 211 | 2.4% |
| Country | Australia | 561 | 6.4% |
| | Egypt | 31 | 0.4% |
| | Italy | 329 | 3.7% |
| | Kenya | 243 | 2.8% |
| | Lebanon | 54 | 0.6% |
| | Luxembourg | 568 | 6.4% |
| | Malaysia | 499 | 5.7% |
| | Mexico | 1,673 | 19.0% |
| | Moldova | 244 | 2.8% |
| | Nigeria | 231 | 2.6% |
| | Portugal | 3,323 | 37.6% |
| | Singapore | 566 | 6.4% |
| | Spain | 295 | 3.3% |
| | Uganda | 212 | 2.4% |
| Area of Residence*** | Rural | 1,762 | 20.0% |
| | Urban | 6,185 | 70.2% |
| | Other | 809 | 9.2% |
| | NA | 50 | 0.6% |
| No. of Children$ | 0 | 5,664 | 64.2% |
| | 1 | 1,238 | 14.0% |
| | 2 | 1,243 | 14.1% |
| | 3 | 454 | 5.1% |
| | 4 | 160 | 1.8% |
| | >4 | 68 | 0.8% |
| Education Completed§§ | No formal education | 17 | 0.2% |
| | Primary Education | 38 | 0.4% |
| | Secondary Education | 1,100 | 12.5% |
| | College or University | 7,108 | 80.9% |
| | Other | 529 | 6.0% |

*(Continued)*

**Table 1.** (Continued)

| Socio-Demographics | | Frequency (N = 8,829) | Percentage |
|---|---|---|---|
| Religion§§§ | Christian | 3,586 | 41.3% |
| | Muslim | 543 | 6.3% |
| | Buddhist | 311 | 3.6% |
| | Hindu | 156 | 1.8% |
| | Jewish | 20 | 0.2% |
| | Other | 617 | 7.1% |
| | No religion | 2,940 | 33.9% |
| | NA | 510 | 5.9% |

*Age has 245 missing values

** Gender has 35 missing values

*** Area of residence has 23 missing values

§ No of Children has 2 missing values

§§ Education has 100 missing values

§§§ Religion has 146 missing values

## Child marriage

Fourteen countries reported on the occurrence of child marriage. These include Australia, Egypt, Italy, Kenya, Lebanon, Luxembourg, Malaysia, Mexico, Moldova, Nigeria, Portugal, Singapore, Spain, and Uganda. In total, 1,429 (13%) out of 8,829 participants reported that child marriage occurred in their community. Mexico reported the highest occurrence of child marriage 46.4% (663/1429), followed by Kenya 10.6% (152/1429), Uganda 9.6% (137/1429), Portugal 6.3% (90/1429) and Singapore 5.9% (85/1429).

Only 28 (7%) participants expressed intent to organise a marriage for their child. Out of this, 23 participants were from Singapore. With reference to the impact of COVID-19 on plans

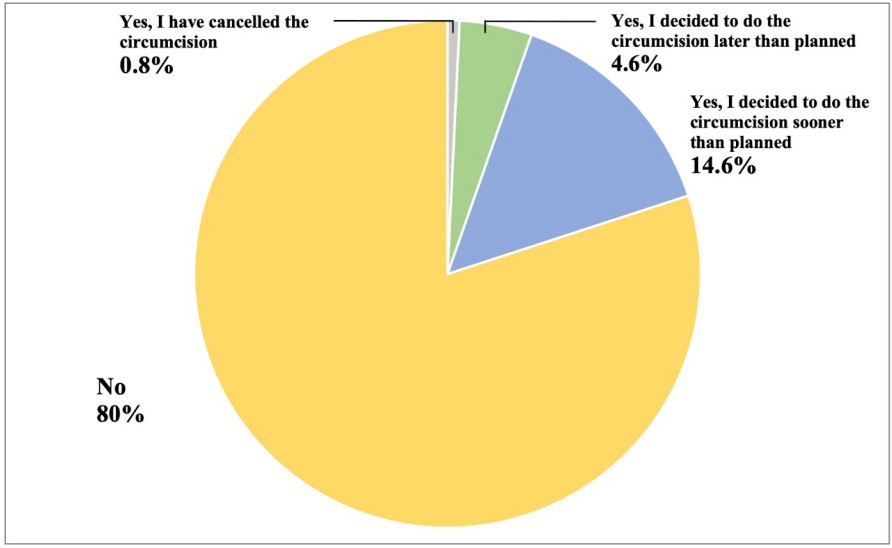

**Fig 1. Frequency of changing plans for organising FGM/C due to COVID-19 (N = 132).** This chart shows the proportion of participants who responded to the survey question "Did the COVID-19 situation change your plans to circumcise your daughter?".

**Table 2. Correlates of changing plans for FGM/C among I-SHARE participants in 12 countries, 2020–2021 (N = 123).**

| | Change in Plans due to COVID-19 (%) | | | | Chi-squared value ($X^2$) | P value |
|---|---|---|---|---|---|---|
| | Cancelled circumcision | Later than Planned | Sooner than Planned | No change | | |
| Change in Economic Situation | | | | | | |
| Worse | 1 (1.9%) | 1 (1.9%) | 3 (5.7%) | 48 (90.6%) | 11.58 | p = 0.07 |
| Better | 0 (0.0%) | 0 (0.0%) | 2 (33.3%) | 4 (66.7%) | | |
| No change | 0 (0.0%) | 5 (7.8%) | 14 (21.8%) | 45 (70.3%) | | |
| Worried about Financial Situation | | | | | | |
| More worried | 0 (0.0%) | 2 (3.9%) | 6 (11.8%) | 43 (84.3%) | 14.03 | p = 0.02 |
| Less worried | 0 (0.0%) | 3 (18.8%) | 5 (31.3) | 8 (50.0%) | | |
| No change | 1 (1.7%) | 1 (1.2%) | 8 (13.8%) | 48 (82.8%) | | |
| Religion | | | | | | |
| Christian | 1 (1.8%) | 3 (5.3%) | 6 (10.5%) | 47 (82.5%) | 23.77 | p = 0.02 |
| Muslim | 0 (0.0%) | 0 (0.0%) | 3 (20.0%) | 12 (80%) | | |
| Buddhist | 0 (0.0%) | 2 (28.6%) | 4 (57.1%) | 1 (14.9%) | | |
| No religion | 0 (0.0%) | 1 (2.6%) | 5 (12.8%) | 33 (84.6%) | | |
| Other | 0 (0.0%) | 0 (0.0%) | 1 (25.0%) | 3 (75%) | | |
| Education Completed | | | | | | |
| No formal education | 0 (0.0%) | 0 (0.0%) | 0 (0.0%) | 1 (100%) | 4.77 | p = 0.97 |
| Primary Education | 0 (0.0%) | 0 (0.0%) | 2 (40.0%) | 3 (60.0%) | | |
| Secondary Education | 0 (0.0%) | 0 (0.0%) | 3 (18.8%) | 13 (81.3%) | | |
| College or University | 1 (1.0%) | 6 (6.0%) | 14 (14.0%) | 79 (79.0%) | | |

to arrange child marriage, 13.5% of participants stated the pandemic had affected their decision to arrange child marriage. About 7.7% (29/384) of participants stated that the pandemic had resulted in them arranging a marriage sooner than planned, while 5.3% (20/384) stated later than planned (Fig 2); less than 1% (3/384) of participants cancelled marriage plans.

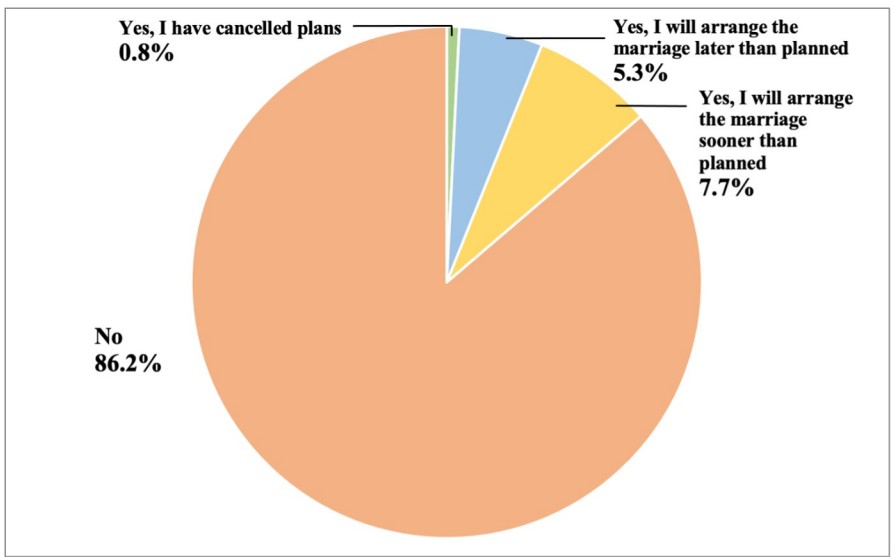

**Fig 2. Frequency of changing plans for arranging child marriage due to COVID-19 (N = 384).** This chart shows the proportion of participants who responded to the survey question "Did the COVID-19 situation change your plans to arrange a marriage for your adolescent child(ren)?".

Univariate associations of the correlates of change in plans to arrange child marriage due to COVID-19 (Table 3) showed that those who experienced a loss of income (partial or total loss) were more likely to advance plans to arrange child marriage ($X^2$ = 20.76; p = 0.002), with 25% of people reporting total loss income expressing intent to arrange the marriage sooner than planned. However, our results also found that 67% of participants reporting a total loss and 91% reporting partial loss did not change plans to arrange marriage. Worry about financial situation showed a negative change in decision to change plans ($X^2$ = 33.17; p<0.001), with those less worried arranging child marriage sooner than planned. Interestingly, 92% of those who were more worried about their financial situation did not change plans to arrange a child marriage. Education was also found to be significantly associated with changing plans ($X^2$ = 30.52; p = 0.002), with participants completing secondary and college education having advanced or delayed plans to marry their child. However, a large proportion of these participants also reported no change on their plans to arrange a marriage. Further, correlates such as religion ($X^2$ = 27.91; p = 0.06) and change in family composition ($X^2$ = 1.71; p = 0.64) showed non-significant associations. No association was found between change in plans and other socioeconomic factors such as area of residence. Further, 57% (733/1284) of participants stated that COVID-19 put adolescents at an increased risk of early marriage (S2 Fig). Note that participant's perception of risk was not found to be associated with the decision to change plans to arrange child marriage.

**Table 3. Univariate association between change in childhood marriage plans within the I-SHARE survey in 14 countries, 2020–2021 (N = 384).**

|  | Change in Plans due to COVID-19 (%) | | | | Chi-squared value ($X^2$) | P value |
|---|---|---|---|---|---|---|
|  | Cancelled Marriage | Later than Planned | Sooner than Planned | No change |  |  |
| Change in Income |  |  |  |  |  |  |
| Total Loss of Income | 1 (3.6%) | 1 (3.6%) | 7 (25.0%) | 19 (67.9%) | 20.76 | p = 0.002 |
| Partial loss of income | 1 (0.76%) | 7 (5.3%) | 3 (2.3%) | 120 (91.6%) |  |  |
| No loss of income | 1 (0.63%) | 8 (5.0%) | 13 (8.2%) | 137 (86.2%) |  |  |
| Worried about Financial Situation |  |  |  |  |  |  |
| More worried | 1 (0.6%) | 8 (4.5%) | 5 (2.8%) | 163 (92.1%) | 33.17 | p<0.001 |
| Less worried | 1 (2.86%) | 6 (17.1) | 8 (22.9%) | 20 (57.1%) |  |  |
| No change | 1 (0.7%) | 5 (3.6%) | 16 (11.6%) | 116 (84.1%) |  |  |
| Education Completed |  |  |  |  |  |  |
| No formal education | 0 (0.0%) | 0 (0.0%) | 0 (0.0%) | 1 (100%) | 30.52 | p = 0.002 |
| Primary Education | 0 (0.0%) | 0 (0.0%) | 4 (44.4%) | 5 (55.6%) |  |  |
| Secondary Education | 0 (0.0%) | 7 (12.7%) | 8 (14.6%) | 40 (72.7%) |  |  |
| College or University | 3 (4.8%) | 12 (19.0%) | 16 (25.4%) | 32 (50.8%) |  |  |
| Religion |  |  |  |  |  |  |
| Christian | 1 (0.5%) | 10 (5.2%) | 9 (4.7%) | 172 (89.6%) | 27.91 | p = 0.06 |
| Muslim | 0 (0.0%) | 2 (9.1%) | 2 (9.1%) | 18 (81.8%) |  |  |
| Buddhist | 0 (0.0%) | 3 (13.6%) | 6 (27.3%) | 13 (59.1%) |  |  |
| Hindu | 0 (0.0%) | 0 (0.0%) | 0 (0.0%) | 5 (100%) |  |  |
| Jewish | 0 (0.0%) | 0 (0.0%) | 0 (0.0%) | 1 (100%) |  |  |
| No religion | 1 (1.4%) | 2 (2.8%) | 6 (8.5%) | 62 (87.3%) |  |  |
| Other | 1 (3.2%) | 2 (6.5%) | 6 (19.4%) | 22 (71.0%) |  |  |
| Change in Family Composition due to COVID-19 |  |  |  |  |  |  |
| Change in composition | 0 (0.0%) | 4 (7.0%) | 3 (5.3%) | 50 (87.5%) | 1.71 | p = 0.64 |
| No change in composition | 3 (1.0%) | 15 (5.6%) | 26 (8.9%) | 247 (84.9%) |  |  |

## Discussion

Our study shows that harmful cultural practices such as FGM/C and child marriage occurred in several countries during the pandemic. COVID-19 restrictions may have accelerated plans for FGM/C and child marriage. Financial problems such as the loss of income was found to be a potential driver of child marriage. Further, more than half of the participants felt that lockdowns and other COVID-19 measures had increased the risk of FGM/C and child marriage. Our study expands on the literature on harmful practices by using several non-convenience sampling methods, leveraging online methods that can maintain privacy, and administering the survey in multiple countries.

Our study found that FGM/C persisted in nine countries during COVID-19 restrictions. Other studies in the area have highlighted the presence of FGM/C during the pandemic due to lack of access to essential protection and support services and the implementation of legal interventions [19]. About 3% of participants reported that female circumcision was practised within their community during COVID-19 restrictions. For example, one in three participants in Kenya reported on the occurrence of FGM/C, previous studies have shown a high prevalence of female circumcision in Kenya, with over 27% of women aged 15–49 having undergone the practise [32–34]. Our results also showed Singapore reporting the largest proportion of people intending to organise female circumcision. A previous study found that even though Singapore has a high prevalence of FGM/C, it is often unreported as it is only practised within certain ethnic groups and other parts of the population largely unaware of the presence of female circumcision within their society [35].

We found that one in five participants in the fourteen countries reported child marriage as occurring in their community during COVID-19 restrictions. Other literature in this area which highlight the increase in child marriage between the initial lockdown period in 2020, with some countries seeing an increase in the number of calls to child marriage helplines [36, 37]. Further, over two-thirds of Ugandan and Kenyan participants in our study reported that child marriage occurred within their community. This is consistent with a UNICEF report suggesting that Africa had one of the highest rates for child marriage [38]. Singapore had the highest number of participants reporting intent to arrange a child marriage, however, similar to FGM/C there is not a lot of literature to support this.

The COVID-19 pandemic influenced participants' decision to organise FGM/C, with 1 in 7 participants accelerating circumcision because of the pandemic. Previous data which showed that many parents were organising circumcision sooner than planned since girls would have time to recover and their absence from school would not be noticed [15]. Further, financial worry was found to be associated with the decision to advance FGM/C amid the pandemic. Other literature in the area has shown the link between poor financial situation such as poverty and FGM/C [18, 19]. Our study also found that very few participants cancelled plans for FGM/C indicating that the practice was a rite of passage which communities would not want to give up despite the pandemic.

COVID-19 restrictions influenced participants' decision to arrange childhood marriage; approximately 8% of participants stated that the pandemic led them to arrange a marriage sooner than planned. Similar, to FGM/C financial situation such as the loss of income was found to be a driver of accelerated child marriage during the pandemic; 1 in 5 participants that reported total loss income arranged the marriage sooner than planned. Further, many families reporting both partial or total loss of income did not delay or cancel plans for marriage and reported no change in their plans to arrange a marriage. This observation could indicate that marriage was seen as essential despite financial losses due to the pandemic. Previous studies have also found that economic losses and poverty due to COVID-19 to be associated with a

higher risk of childhood marriage. For example, in Indonesia, many parents were marrying their children to secure their future during uncertain times, thus increasing the risk of child marriage [39]. Further, other studies have shown that crises tend to increase the occurrence of harmful cultural practices. During the Ebola outbreak, child marriages in West Africa increased exponentially as many girls ended up getting pregnant due to the closure of schools [40, 41]. Our study showed that more than half of the participants felt that COVID-19 has increased the risk of female circumcision and child marriage. Other studies have also corroborated this finding, with participants in African countries such as Kenya and Uganda perceiving that the pandemic increased cases of both FGM/C and child marriage; economic losses and closure of schools was cited as the main reason for this increase [18].

This study has several limitations. First, both FGM/C and child marriage are sensitive topics, and the survey items were optional for participants to complete. As a result, self-reported intention to organise and conduct FGMC and child marriage may have been biased. Many participants may have opted not to participate in this part of the questionnaire due to the sensitivity of the items. This is particularly possible in countries having strict laws against the two practices. Due to the fear of being stigmatised, many may have chosen to not respond to the questions on these practices. Thus, our data on occurrence are likely an under-estimate. Second, even though the overall sample size for the two practices was large (N = 8,829), the sample sizes for individual survey items were relatively small. Therefore, only limited analysis of the data could be carried out. It was also not possible to detect potential collinearity between the variables. Third, 23 participants out of the 28 that intended to arrange a marriage for their child were from Singapore, which skewed the results to that country's context. Similarly, for the intention to organise circumcision, 11 out of 18 participants were from Singapore. Fourth, our data found that only 8% of participants in Egypt stated that FGM/C occurred within their communities, whereas Egypt has a very high prevalence rate for female circumcision–over 50% in recent birth cohorts and over 90% in older birth cohorts [42]. Last, the data was biased towards urban settings as over 70% of participants were from urban/semi-urban areas. Since the survey was mainly online, people in remote areas with weak internet connection and less online presence may not have been able to access the survey. We know from previous studies that the occurrence of FGM/C and child marriage are mainly concentrated in rural areas. Hence, it is possible that a large portion of people practising may have been missed out.

In addition, this study has several implications. While this analysis was able to showcase the observed impact of COVID-19 restrictions on harmful cultural practices, it was based entirely on cross-sectional data. This was useful in capturing the immediate perceived impact of the pandemic on intention and plans to organise FGM/C and child marriage; however, longitudinal research is needed to assess the long-term and actual impact of COVID-19 on these harmful cultural practices. Due to the limited data for each survey item, it was not possible to assess all the socioeconomic factors driving the decision to delay or advance FGM/C and child marriage; thus, a larger dataset would be more suitable. Since the survey was administered during the first lockdown period when manpower and access was limited, it was also not possible to collect qualitative data on participant's experiences with the two practices. Moreover, the sensitive nature of the two practices prevented us from capturing social aspects and the reasoning behind the participants' decision to change or not change their plans. Thus, future studies could use qualitative interviews, in addition to quantitative analysis, as they are more suitable for exploring sensitive topics [43].

Our study has shown the association between COVID-19 restrictions such as lockdowns and closure of schools and decision to organise FGM/C sooner than planned. This shows the importance of schools in protecting girls from circumcision; however, further studies are required to corroborate this association. The study has also shown that FGM/C and child

marriage have similar contexts with financial worries and economic losses being a contributing factor for both practices. Another study has also shown the link between the two practices, with families suffering from economic losses being forced to sell their young daughters for a bride price and in some cases requiring them to undergo circumcision [19]. Further, there is evidence from previous epidemics that effective policies can result in a suspension of circumcision activities [40, 44]. For instance, the government of Sierra Leone introduced a temporary suspension on female circumcision during the Ebola outbreak, which was found to lower the number of circumcisions during that time period [45].

## Conclusion

This analysis explored the perceived impact of the COVID-19 pandemic on harmful cultural practices of FGM/C and child marriage. Our findings show that many families intended to carry out both practices sooner than planned, while some delayed or even cancelled plans. The financial situation of the family impacted plans to arrange a marriage for their children. We also found that there were fewer cancellations, especially for FGM/C, as the practice was a rite of passage which communities would not want to give up during or as a result of a pandemic. Our study has shown that more work still needs to be done to control and end these harmful cultural practices.

## Supporting information

**S1 Fig. Perception of additional risk of female circumcision due to COVID-19 on girls (N = 278).** This chart shows the proportion of participants who felt the pandemic had increased the risk of FGM/C.
(DOCX)

**S2 Fig. Perception of additional risk of childhood marriage due to COVID-19 on girls (N = 1,284).** This chart shows the proportion of participants who felt the pandemic had increased the risk of child marriage.
(DOCX)

**S1 Table. Survey items relating to FGM/C and child marriage in the I-SHARE survey, 2020–2021.** This table shows the survey items relating to FGM/C and child marriage and the response options for each question.
(DOCX)

## Acknowledgments

I-SHARE, Consortium (Ghent, LSHTM), all participating countries (https://ishare.web.unc.edu/team-members/).

## Author Contributions

**Conceptualization:** Marleen Temmerman, Tammary Esho, Samuel Kimani, Joyce Omwoha, Joseph D. Tucker.

**Data curation:** Marleen Temmerman, Tammary Esho, Samuel Kimani, Joyce Omwoha.

**Formal analysis:** Shania Pande, Joseph D. Tucker.

**Investigation:** Marleen Temmerman, Joseph D. Tucker.

**Methodology:** Tammary Esho, Samuel Kimani, Joyce Omwoha.

**Supervision:** Eneyi E. Kpokiri, Joseph D. Tucker.

**Writing – original draft:** Shania Pande, Eneyi E. Kpokiri.

**Writing – review & editing:** Shania Pande, Simukai Shamu, Amr Abdelhamed, James Munyao Kingoo, Sarah Van de Velde, Eneyi E. Kpokiri, Joseph D. Tucker.

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
