## [Decision Letter · Decision Letter 0]

3 Aug 2023

PONE-D-23-10482COVID-19 and Female Genital Mutilation/Cutting and Child Marriage: An online multi-country cross sectional surveyPLOS ONE

Dear Dr. Kpokiri,

Thank you for submitting your manuscript to PLOS ONE. After careful consideration, we feel that it has merit but does not fully meet PLOS ONE’s publication criteria as it currently stands. Therefore, we invite you to submit a revised version of the manuscript that addresses the points raised during the review process.

We look forward to receiving your revised manuscript.

Kind regards,

Zeus Aranda, MSc MBA

Academic Editor

PLOS ONE

Journal Requirements:

2. Please note that in order to use the direct billing option the corresponding author must be affiliated with the chosen institute. Please either amend your manuscript to change the affiliation or corresponding author, or email us at plosone@plos.org with a request to remove this option.

3.In your Data Availability statement, you have not specified where the minimal data set underlying the results described in your manuscript can be found. PLOS defines a study's minimal data set as the underlying data used to reach the conclusions drawn in the manuscript and any additional data required to replicate the reported study findings in their entirety. All PLOS journals require that the minimal data set be made fully available. For more information about our data policy, please see http://journals.plos.org/plosone/s/data-availability.

Reviewers' comments:

Reviewer's Responses to Questions

**Comments to the Author**

1. Is the manuscript technically sound, and do the data support the conclusions?

Reviewer #1: Yes

Reviewer #2: Yes

Reviewer #3: No

2. Has the statistical analysis been performed appropriately and rigorously? 

Reviewer #1: Yes

Reviewer #2: Yes

Reviewer #3: N/A

3. Have the authors made all data underlying the findings in their manuscript fully available?

Reviewer #1: No

Reviewer #2: Yes

Reviewer #3: No

4. Is the manuscript presented in an intelligible fashion and written in standard English?

Reviewer #1: Yes

Reviewer #2: Yes

Reviewer #3: Yes

5. Review Comments to the Author

Reviewer #1: Overall a very nice article, interesting study on a relevant topic. The online multi-country cross sectional survey show that in times of COVID 19 both the cases of female genital mutilation and child marriage went up, and the authors conclude as it has these results are based on self-reporting, probably the real figures are higher. Some issues are only presented in the discussion and it is recommended to also integrate them in the results. Link to laws should be taken out.

Some specific comments

Abstract overall informative !

p.1

-Abstract starts with harmful cultural practices and it is maybe better to start with Female genital mutilation etc

- the analysis assessed the impact of COVID-19 restrictions-good already here to present the definition or give examples here.

p.2

- can you give a bit more detail about “ sooner than planned”

- some repetition in the abstract- sooner than plan

- why would strict enforcement of regulations discourage these practices? For both practices there are assumed big rituals at community level, but is this really the case?

Introduction

p. 2

- the substantial economic costs are referenced with the website of WHO is there is no date, please add undated or accessed when. Same for other references

- FGM/C be performed in preparation for child marriage – please rephrase as marriageability was linked to it indeed but in some countries less the case nowadays (CF Kenya) and sometimes first leading to sexual activity and pregnancy.

p.3 Bangladesh reported an increased number of child marriages but studies from before the COVIC period!

p.3 parents to arrange a marriage in time of conflict to protect them from harm? What harm?

Methods and Data Collection are fine but good to explain why there was no option to add a qualitative component to the study.

P. 4 here it is mentioned that FGM/C and child marriage were optional components and included in 14 countries this clarifies why 14 countries reported on it as mentioned in the abstract

p. 5 Table 1; normally you would think a table as this would be better placed at the end of the article but here I think it helps to understand the findings; however maybe both table 1,2,3 at the end?

p.7 Good to give some detailed information why both change in economic situation and educational level were found non-significant.

p.8 why do the authors suspect that so many -23 participants were from Singapore and not from other South Asian countries?

Discussion

p.9 financial problems- do we know more?

p.10 some information about the occurrence of FGM/C in Kenya and Singapore should be better presented in the Results section and only discussed here?

- sooner than planned with reference to Orchid please provide more details

- 1 out of 5 – seems not a high number – how do you explain

p. 11

- the link to study of Esho at all 2022 is fine here, it underlines your findings

- the only 8% of Egypt is indeed surprising – bias to urban settings and becase the syudy was online could also be mentioned in the abstract?

p.12

- link to closing of schools fine, but also maybe already ,mention this in the results and come back to it here?

p. 12 the importance of more stringent laws are not related to this article and the results and should be taken out!! That is a whole other discussion

Reviewer #2: It is important to comprehend how the pandemic has affected harmful practices such as FGMC/C and child marriage. The only comments I have are as follows:

Did COVID-related financial difficulties have the biggest impact on changes in rates of child marriage and genital mutilation? Or were there other COVID-related restrictions that also had an effect?

According to the conclusions, there is a claim that the decision to advance FGM/C during the pandemic is associated with the family's financial situation. However, this claim was not found to be statistically significant.( p=0.07)

Reviewer #3: 1. Is the manuscript technically sound, and do the data support the conclusions? No

The authors make overly broad statements and cite a single reference to support it, which often is not a peer-reviewed journal article.

The authors make misleading statements that their own sources and their own contradict, or at least do not support. One example (among many): “FGM/C may be performed in preparation for child marriage.” Yet the UNICEF report they cite states that “In countries affected by both practices, most women experience neither practice or only one of the two.” Moreover, their own data contradict the statement; only in Portugal does prevalence of FGC correlate with prevalence of child marriage.

A second, more egregious, example: their data shows that 7.7% of participants said they would arrange a marriage for their daughters sooner than planned, 5.3% later than planned, and 1% cancelled marriage plans. Yet in their conclusion, the authors state only that “The findings show that the pandemic has resulted in many families carrying out both practices sooner than planned.” They do not mention that almost as many families postponed or canceled the marriage.

The survey respondents—mostly young, childless, college-educated, urban “participants” associated with a research network—do not represent people who actually perform FGC or child marriage. Almost 60% of the participants are from Mexico (19%) and Portugal (37.6%), out of 14 countries. Mexico has a high prevalence of child marriage but no FGC; Portugal is single digits for both (I assume because of immigrant groups).

Poor presentation of descriptive statistics. One example (of many): “About 13.6% (18/132) of participants intented [sic] to organise circumcision for their daughter.” But there is no explanation of where the 132 denominator comes from. This follows: “Out of these, 19(14.6%) participants intended to circumcise their daughters sooner than planned, while 6 (4.6%) stated later than planned, only one (0.8%) participant cancelled plans for circumcision.” It looks like 18 became 19+6+1.

The authors included opinion questions from “participants” who likely have no real connection with the practices involved, especially FGC.

The questions about planning FCG or child marriage are hypothetical, about intentions and plans, not about real events. Without follow-up, there is no way to conclude that COVID really affected either of the practices.

The authors include two paragraphs (the “implications” paragraph is actual a continuation of limitations) that describe—very well—several limitations, which should have precluded making the conclusions they did and submitting the manuscript for publication.

*2. Has the statistical analysis been performed appropriately and rigorously? No

The numbers of “participants” who actually carry out the practices being investigated are so small that it behooves the authors to show that their sample size is adequate for their results to be significant.

One example (among many): Of the 7,234 participants (from twelve countries), 249 (3.4%) said FGC occurs in their “community”, and of these (??), 18 said they intended to organize FGC for their daughter. It is preposterous to think that 18 respondents can lead to a significant finding.

Another example (from Discussion section): “The COVID-19 pandemic influenced participants’ decision to organize FGM/C, with 1 in 7 participants accelerating circumcision because of the pandemic.” But that statement is based on only 19 (14.6%) participants—not to mention that 6 (4.6%) planned to delay FGC and 1 (0.8%) planned to cancel.

*3. Have the authors made all data underlying the findings in their manuscript fully available? No

The supporting materials seem to be incomplete. In particular, the authors state in their discussion section that “Our study has showen [sic] the association between closure of schools due to the pandemic and decision to organize FGC/M sooner than planned.” Yet nowhere is there any mention of school closures in the survey questions or data description.

*4. Is the manuscript presented in an intelligible fashion and written in standard English? yes

The writing for the most part is fine, but there are many typos.

General:

FGC covers a wide spectrum of practices, from full infibulation to a mere symbolic prick or scraping. The authors narrow the definition from the source they themselves cite to “the partial or complete removal of the female genitalia...”

The authors seem to be trying to claim that human rights (in this case, girl’s rights) have been negatively impacted by COVID. But they cannot make that claim based on the data they present here.

6. PLOS authors have the option to publish the peer review history of their article (what does this mean?). If published, this will include your full peer review and any attached files.

Reviewer #1: No

Reviewer #2: **Yes: **Mercedes Alarcon

Reviewer #3: No

---

## [Author Response · Author response to Decision Letter 0]

4 Oct 2023

Reviewer #1: Overall a very nice article, interesting study on a relevant topic. The online multi-country cross sectional survey show that in times of COVID 19 both the cases of female genital mutilation and child marriage went up, and the authors conclude as it has these results are based on self-reporting, probably the real figures are higher. Some issues are only presented in the discussion, and it is recommended to also integrate them in the results. Link to laws should be taken out.

Response: The manuscript has been updated to include some issues in the results sections and the link to the laws has been taken out.

Some specific comments

Abstract overall informative !

p.1

-Abstract starts with harmful cultural practices, and it is maybe better to start with Female genital mutilation etc

- the analysis assessed the impact of COVID-19 restrictions-good already here to present the definition or give examples here.

Response: The abstract has been updated to start with Female Genital Mutilation and Child Marriage. The definition of COVID-19 restrictions has been included in the introduction.

p.2

- can you give a bit more detail about “ sooner than planned”

Response: Sooner than planned refers to participants intending to arrange circumcision or child marriage earlier than they had planned to because of the pandemic. More details about the survey items can be found in the supporting information.

- some repetition in the abstract- sooner than plan

Response: This has been revised and now reads “planned to organise FGM/C earlier.”

- why would strict enforcement of regulations discourage these practices? For both practices there are assumed big rituals at community level, but is this really the case?

Response: While these are still practiced as cultural rituals in many local communities, we envisage strict enforcement of regulations discouraging these practices will potentially reduce occurrences.

Introduction

p. 2

- the substantial economic costs are referenced with the website of WHO is there is no date, please add undated or accessed when. Same for other references

Response: The date for references has been added.

- FGM/C be performed in preparation for child marriage – please rephrase as marriageability was linked to it indeed but in some countries less the case nowadays (CF Kenya) and sometimes first leading to sexual activity and pregnancy.

Response: This has been revised and now says “FGM/C may be performed in preparation for child marriage; however, this has reduced in past few years. In some settings, it is performed to prepare young girls for sexual activity and pregnancy.”

p.3 Bangladesh reported an increased number of child marriages but studies from before the COVIC period!

Response: This has been revised and now reads “Bangladesh reported an increased number of child marriages prior to and during COVID-19”

p.3 parents to arrange a marriage in time of conflict to protect them from harm? What harm?

Response: This harm includes preventing the child from being kidnapped or sexually assaulted during periods of conflict. This has been included in the revised manuscript.

Methods and Data Collection are fine but good to explain why there was no option to add a qualitative component to the study.

Response: This study was conducted as a multi-country study during and just after the COVID period with restrictions still in place in many countries. The sensitive nature of the topic and the fact that it is a criminal practice in some countries make it difficult to organize in-person data collection processes such as in-depth interviews. Finally, a lot of resources and time would be needed to translate the survey instrument into all the local language and pilot to ensure meaning and consistency is maintained across all the countries/languages. This has now been addressed in the limitations section.

P. 4 here it is mentioned that FGM/C and child marriage were optional components and included in 14 countries this clarifies why 14 countries reported on it as mentioned in the abstract

p. 5 Table 1; normally you would think a table as this would be better placed at the end of the article but here I think it helps to understand the findings; however maybe both table 1,2,3 at the end?

Response: Table 1, 2, and 3 present demographic data and thee results from the chi-squares tests for the correlates of FGM/C and child marriage that we think helps with the flow and understanding of the results presented and discussed. Hence, we have included in the main body of the results section

p.7 Good to give some detailed information why both change in economic situation and educational level were found non-significant.

Response: From our data, these variables did not have any correlation with intentions and actual practice of female genital cutting or child marriage

p.8 why do the authors suspect that so many -23 participants were from Singapore and not from other South Asian countries?

Response: In Singapore, there are a higher proportion of Malay Muslims, where FGM/C is practised. 

Discussion

p.9 financial problems- do we know more?

Response: The financial problems referred to are the total or partial loss of income or worry about finances. 

p.10 some information about the occurrence of FGM/C in Kenya and Singapore should be better presented in the Results section and only discussed here?

- sooner than planned with reference to Orchid please provide more details

- 1 out of 5 – seems not a high number – how do you explain

Response: The results section has been corrected to include findings from Kenya and Singapore. The Orchid reference has been removed

The lower number could be because of the sensitivity of FGM/C and the participants may not have felt comfortable disclosing their status through an online survey given that it is illegal in many countries. We have addressed this in the limitations.

p. 11

- the link to study of Esho at all 2022 is fine here, it underlines your findings

- the only 8% of Egypt is indeed surprising – bias to urban settings and because the study was online could also be mentioned in the abstract?

Response: This has now been captured in the abstract and reads: “Given it was an online survey, we had more responses from urban areas”

p.12

- link to closing of schools fine, but also maybe already ,mention this in the results and come back to it here?

p. 12 the importance of more stringent laws are not related to this article and the results and should be taken out!! That is a whole other discussion

Response: The link to laws has been taken out.

Reviewer #2: It is important to comprehend how the pandemic has affected harmful practices such as FGMC/C and child marriage. The only comments I have are as follows:

Did COVID-related financial difficulties have the biggest impact on changes in rates of child marriage and genital mutilation? Or were there other COVID-related restrictions that also had an effect?

Response: Based on our survey questions, financial difficulties seemed to have a biggest impact. COVID-19 restrictions such as lockdowns and closure of schools have also impacted the two practises; however, our survey did not include specific questions about the different types of restrictions. 

According to the conclusions, there is a claim that the decision to advance FGM/C during the pandemic is associated with the family's financial situation. However, this claim was not found to be statistically significant.( p=0.07)

Response: This has been amended to say financial worry which was significant rather than financial situation.

Reviewer #3: 1. Is the manuscript technically sound, and do the data support the conclusions? No

The authors make overly broad statements and cite a single reference to support it, which often is not a peer-reviewed journal article.

The authors make misleading statements that their own sources and their own contradict, or at least do not support. One example (among many): “FGM/C may be performed in preparation for child marriage.” Yet the UNICEF report they cite states that “In countries affected by both practices, most women experience neither practice or only one of the two.” Moreover, their own data contradict the statement; only in Portugal does prevalence of FGC correlate with prevalence of child marriage.

A second, more egregious, example: their data shows that 7.7% of participants said they would arrange a marriage for their daughters sooner than planned, 5.3% later than planned, and 1% cancelled marriage plans. Yet in their conclusion, the authors state only that “The findings show that the pandemic has resulted in many families carrying out both practices sooner than planned.” They do not mention that almost as many families postponed or cancelled the marriage.

Response: We have taken in your feedback to remove any contradictory statements. We have also now mentioned that families postponed and cancelled marriage.

The survey respondents—mostly young, childless, college-educated, urban “participants” associated with a research network—do not represent people who actually perform FGC or child marriage. Almost 60% of the participants are from Mexico (19%) and Portugal (37.6%), out of 14 countries. Mexico has a high prevalence of child marriage but no FGC; Portugal is single digits for both (I assume because of immigrant groups).

Response: This has been captured as part of the study limitations

Poor presentation of descriptive statistics. One example (of many): “About 13.6% (18/132) of participants intented [sic] to organise circumcision for their daughter.” But there is no explanation of where the 132 denominator comes from. This follows: “Out of these, 19(14.6%) participants intended to circumcise their daughters sooner than planned, while 6 (4.6%) stated later than planned, only one (0.8%) participant cancelled plans for circumcision.” It looks like 18 became 19+6+1.

Response: We have made it clear where we got the denominator from.

The authors included opinion questions from “participants” who likely have no real connection with the practices involved, especially FGC.

The questions about planning FCG or child marriage are hypothetical, about intentions and plans, not about real events. Without follow-up, there is no way to conclude that COVID really affected either of the practices.

Response: Yes we agree. Hence, in the implications we suggest some follow up work that needs to be carried out to come to more definitive conclusions.

The authors include two paragraphs (the “implications” paragraph is actual a continuation of limitations) that describe—very well—several limitations, which should have precluded making the conclusions they did and submitting the manuscript for publication.

*2. Has the statistical analysis been performed appropriately and rigorously? No

The numbers of “participants” who actually carry out the practices being investigated are so small that it behooves the authors to show that their sample size is adequate for their results to be significant.

One example (among many): Of the 7,234 participants (from twelve countries), 249 (3.4%) said FGC occurs in their “community”, and of these (??), 18 said they intended to organize FGC for their daughter. It is preposterous to think that 18 respondents can lead to a significant finding.

Another example (from Discussion section): “The COVID-19 pandemic influenced participants’ decision to organize FGM/C, with 1 in 7 participants accelerating circumcision because of the pandemic.” But that statement is based on only 19 (14.6%) participants—not to mention that 6 (4.6%) planned to delay FGC and 1 (0.8%) planned to cancel.

Response: We agree that the sample size is too small to make any conclusions. Hence, we are just reporting on our findings.

*3. Have the authors made all data underlying the findings in their manuscript fully available? No

The supporting materials seem to be incomplete. In particular, the authors state in their discussion section that “Our study has showe [sic] the association between closure of schools due to the pandemic and decision to organize FGC/M sooner than planned.” Yet nowhere is there any mention of school closures in the survey questions or data description.

Response: Our study looked at overall COVID-19 restrictions to show the impact on harmful cultural practises and we did not include specific questions about the restrictions.

*4. Is the manuscript presented in an intelligible fashion and written in standard English? yes

The writing for the most part is fine, but there are many typos.

Response: The typos have been corrected.

General:

FGC covers a wide spectrum of practices, from full infibulation to a mere symbolic prick or scraping. The authors narrow the definition from the source they themselves cite to “the partial or complete removal of the female genitalia...”

Response: We have included the full definition in the revised manuscript.

The authors seem to be trying to claim that human rights (in this case, girl’s rights) have been negatively impacted by COVID. But they cannot make that claim based on the data they present here.

Response: We have revised the manuscript to remove this out.

---

## [Decision Letter · Decision Letter 1]

4 Mar 2024

PONE-D-23-10482R1COVID-19 and Female Genital Mutilation/Cutting and Child Marriage: An online multi-country cross sectional surveyPLOS ONE

Dear Dr. Kpokiri,

Thank you for submitting your manuscript to PLOS ONE. After careful consideration, we feel that it has merit but does not fully meet PLOS ONE’s publication criteria as it currently stands. Therefore, we invite you to submit a revised version of the manuscript that addresses the points raised during the review process.

We look forward to receiving your revised manuscript.

Kind regards,

Zeus Aranda, MSc MBA

Academic Editor

PLOS ONE

Journal Requirements:

Reviewers' comments:

Reviewer's Responses to Questions

**Comments to the Author**

1. If the authors have adequately addressed your comments raised in a previous round of review and you feel that this manuscript is now acceptable for publication, you may indicate that here to bypass the “Comments to the Author” section, enter your conflict of interest statement in the “Confidential to Editor” section, and submit your "Accept" recommendation.

Reviewer #2: All comments have been addressed

2. Is the manuscript technically sound, and do the data support the conclusions?

Reviewer #2: Partly

3. Has the statistical analysis been performed appropriately and rigorously? 

Reviewer #2: Yes

4. Have the authors made all data underlying the findings in their manuscript fully available?

Reviewer #2: Yes

5. Is the manuscript presented in an intelligible fashion and written in standard English?

Reviewer #2: Yes

6. Review Comments to the Author

Reviewer #2: Overall the appropiate changes were made , the limitations of the study are clear and well explained.

It would be helpful to explain that the FMG C and child marriage were optional in the survey and that the survey did not include questions about the type of COVID restrictions.

7. PLOS authors have the option to publish the peer review history of their article (what does this mean?). If published, this will include your full peer review and any attached files.

Reviewer #2: **Yes: **Mercedes Alarcon

---

## [Author Response · Author response to Decision Letter 1]

27 Apr 2024

Reviewer #2: Overall the appropriate changes were made, the limitations of the study are clear and well explained.

It would be helpful to explain that the FMG C and child marriage were optional in the survey and that the survey did not include questions about the type of COVID restrictions.

Response: We have added this clarification in the methods section under the paragraph on survey items. The las two sentences now reads as follows: “Note that the survey items for FGM/C and child marriage were included as optional components of the main survey and 14 countries included these two components in the survey. The survey did not include questions about the type of COVID restrictions”. 

Reviewer #2: Overall the appropriate changes were made, the limitations of the study are clear and well explained.

It would be helpful to explain that the FMG C and child marriage were optional in the survey and that the survey did not include questions about the type of COVID restrictions.

Response: We have added this clarification in the methods section under the paragraph on survey items. The las two sentences now reads as follows: “Note that the survey items for FGM/C and child marriage were included as optional components of the main survey and 14 countries included these two components in the survey. The survey did not include questions about the type of COVID restrictions”.

---

## [Editor Report · Decision Letter 2]

16 May 2024

COVID-19 and Female Genital Mutilation/Cutting and Child Marriage: An online multi-country cross sectional survey

PONE-D-23-10482R2

Dear Dr. Kpokiri,

We’re pleased to inform you that your manuscript has been judged scientifically suitable for publication and will be formally accepted for publication once it meets all outstanding technical requirements.

Kind regards,

Zeus Aranda, MSc MBA

Academic Editor

PLOS ONE
---

## [Editor Report · Acceptance letter]

21 Oct 2024

PONE-D-23-10482R2 

PLOS ONE

Dear Dr. Kpokiri, 

I'm pleased to inform you that your manuscript has been deemed suitable for publication in PLOS ONE. Congratulations! Your manuscript is now being handed over to our production team.

Kind regards, 

on behalf of

Mr. Zeus Aranda 

Academic Editor

PLOS ONE